# Changes in Physical Activity during the COVID-19 Pandemic—An Analysis of Differences Based on Mitigation Policies and Incidence Values in the Federal States of Germany

**DOI:** 10.3390/sports9070102

**Published:** 2021-07-15

**Authors:** Franziska Beck, Michael Mutz, Eliane Stephanie Engels, Anne Kerstin Reimers

**Affiliations:** 1Department of Sport Science and Sport, Friedrich-Alexander-University Erlangen-Nuremberg, Gebbertstrasse 123b, 91058 Erlangen, Germany; franzi.beck@fau.de (F.B.); eliane.engels@fau.de (E.S.E.); 2Institute of Sport Science, Justus-Liebig-University Gießen, Kugelberg 62, 35394 Giessen, Germany; michael.mutz@sport.uni-giessen.de

**Keywords:** physical activity, COVID-19, mitigation policies, incidence values, changes

## Abstract

Measures to slow down the spread of coronavirus SARS-CoV-2 have had an impact on the daily life and physical activity (PA) of many people. Nevertheless, in Germany, mitigation policies and incidence values vary widely across the federal states (Länder). Thus, the aim of the present study was to investigate regional differences in PA during the coronavirus pandemic. This study is based on the SPOVID project (Examining physical activity and sports behavior in the face of COVID-19 pandemic: a social inequality perspective) that incorporates a large-scale, representative cross-sectional survey representing the German population (≥14 years). Based on the survey that took place in October 2020 (*N* = 1477), we investigated the relationships between the COVID-19 incidence values as well as the mitigation policies across the federal states in Germany and changes in PA. Pearson correlations indicated a strong negative relationship between PA change and 7-day incidence values (r = −0.688 **, *p* = 0.009) and a moderate negative relationship between PA changes and an index of mitigation policies (r = −0.444, *p* = 0.112). Higher 7-day incidence values and stricter mitigation policies were associated with a stronger decline in PA levels. Therefore, it is important to support people to stay active even if there are restrictions. In particular, in federal states and regions with high incidences and stricter mitigation policies, measures to promote health-enhancing PA are necessary.

## 1. Introduction

In general, physical activity (PA) is associated with numerous health benefits [1,2,3]. Several reviews revealed positive associations between PA and reduced risk of all-cause mortality and several chronic diseases such as cardiovascular diseases, cancer incidence and mortality, as well as type 2 diabetes and obesity [1,2,4]. Besides the physiological health benefits, there are also positive associations of PA with mental health. In particular, a study has shown that even a short period of one week, in which formerly active people were excluded from sports and PA, has deleterious effects on depression and mood [5]. Reviews indicated an increase in anxiety and depression [2,6,7] and lower life satisfaction [8,9] caused by a decrease in PA. Additionally, individuals who regularly exercise show lower rates of acute respiratory infection incidence, duration and intensity of symptoms and risk of mortality from infectious respiratory diseases [10,11,12,13]. However, modifications to the immune response depend on the intensity, duration and type of exercise [10]. Regularly executing PA at an appropriate moderate level of intensity improves the immune system by exerting immunoregulatory effects, controlling the viral gateway and modulating inflammation and can have an effect on the innate and adaptive immune response [14]. These factors may also help in protecting individuals against infections such as COVID-19. Additionally, the benefits of regular PA include immunovigilance and improved immune competence, which help to control pathogens [10,15]. Thus, especially during the current COVID-19 pandemic, PA plays an important role as a protective factor against poor mental and physical health as well as in improving resistance against infections [16].

Despite the health benefits of PA, many adults in Germany and worldwide are generally not as sufficiently physically active as World Health Organization (WHO) guidance recommends [17]. As postulated in the WHO PA recommendations, adults should either be moderately physically active for 150 min a week or intensively physically active for 75 min. According to representative data from Germany based on adults aged 18 to 79 years obtained from the “German Health Interview and Examination Survey for Adults” (DEGS1), only 35% of women and 43% of men in Germany achieve the WHO PA recommendations [18].

Since the outbreak of the COVID-19 pandemic, mitigation policies were implemented by governmental institutions, which have had an impact on everyone’s daily life and thus may also have had an impact on levels of PA. Mutz and Gerke [19] found a significant decline in leisure-time sports and exercise in adults from Germany during the lockdown in March/April 2020, but light outdoor activities were a substitute for sports and exercise. Concurrently, with recent data from the cross-sectional COVID-19 Snapshot Monitoring (COSMO) Germany study during the first lockdown in April 2020, Maertl et al. [20] revealed that 42.6% of adults aged 18 and 74 years fulfilled the WHO recommendations. Another study showed a decrease in PA in 44.5% of students of major Bavarian universities, while 32.8% increased their PA from January 2020 (before lockdown) to March 2020 (during the lockdown) [21].

These changes in PA during the course of the COVID-19 pandemic could vary according to regional incidence rates of COVID-19 and mitigation policies implemented by federal states. For example, a study investigating mobility from January till May 2020 in Bavaria, Northrhine-Westphalia, Hamburg and Berlin provided evidence for an association between confirmed COVID-19 cases, reproduction number and individual mobility [22]. Furthermore, a study based on data collected in May 2020 revealed an association between the incidence rates in each federal state in Germany and mental health, with fewer mental health issues in states with lower incidence values [23].

To date, only one study has investigated the relationships of mitigation policies and incidence rates of infections of COVID-19 with changes in PA levels: a Canadian study investigated regional differences in PA in children and adolescents with regard to the number of infection cases in April 2020. Children and adolescents living in areas with the highest prevalence of COVID-19 and the most stringent restrictions on access to outdoor spaces had the greatest decline in time spent outdoors and in outdoor play [24].

Federalism in Germany divides political power between the government and the Länder (federal states) in such a way that both political levels are responsible for certain (constitutionally defined) tasks [25]. Despite regular consultations between the chancellor and the premiers of 16 Länder during the COVID-19 pandemic, the 16 Länder have the legislative power to regulate their mitigation policies by themselves. Hence, the German COVID-19 crisis management has been described as coordinated, but largely decentralized [26].

During the lockdown in March and April 2020 (first wave), strict mitigation policies related to public and private life were imposed across all Länder. However, beginning in May 2020, each state defined its own measures to face the pandemic in a phase of decentralized and unilateral decision making [26]. In fact, the states adopted very different measures and regulations. In particular, some Länder allowed organized sports without any restrictions while others allowed a maximum number of 30 persons. Furthermore, some Länder allowed large events with more than 1000 people whereas others restricted such events to a maximum of 200 persons [27]. Consequently, different patterns of mitigation policies concerning public and private life have been imposed across the country, according to regional infection dynamics [28].

Although it is likely that different mitigation policies and incidence rates affect PA levels, there have been no investigations of changes in PA at the level of Länder to date. Hence, it is unknown if different mitigation policies and incidence rates of COVID-19 are related to changes in adults’ PA behavior during the pandemic. Thus, the aim of our study was to investigate differences in changes in adults’ PA levels across the Länder in Germany and their associations with incidence rates as well as mitigation policies during the COVID-19 pandemic in October 2020.

## 2. Materials and Methods

### 2.1. Study Design

The present exploratory study is part of the SPOVID project “Examining physical activity and sports behavior in the face of COVID-19 pandemic: a social inequality perspective”, which is based on a representative large-scale, cross-sectional survey of the German population (≥14 years). Data collection took place from 16 October to 3 November, 2020 by means of computer-assisted web interviewing. The study was approved by the local Ethics Committee (Ref. No. 387_20B) and was in accordance with the 1964 Declaration of Helsinki. All participants provided written informed consent for study participation.

### 2.2. Procedure and Study Participants

The survey made use of an existing nation-wide online panel (forsa.omninet) to which access was provided by Forsa, a leading organization for public opinion polls. The recruitment for the surveys took place solely offline via telephone interviewing, so that people who rarely use the internet were still represented in the sample. The panel is representative of the German population regarding age, gender, education and place of residence. After giving informed consent to be contacted for the survey, participants received an invitation e-mail with a link to the questionnaire. Participants were able to answer the survey with a tablet, smartphone or computer. The survey took about 10–15 min to complete and contained 40 questions about PA, perception of vulnerability, mental well-being, self-efficacy and sociodemographic data. A total of *N* = 1507 respondents completed the survey.

### 2.3. Measures

Changes in physical activity levels. Participants were asked retrospectively about their PA behavior during the week preceding the survey and during the time before the COVID-19 pandemic. They indicated the hours per week spent on (1) sport and exercise, (2) active travel (by foot or by bicycle), (3) light outdoor activities (like hiking or going for walks) and (4) housework and gardening. Participants used eight answer categories to indicate their time per week spent on each domain (0 = “no time at all”, 0.5 = “less than 1 h”, 1 = “1 h”, 2 = 2 h”, 3.5 = 3 to 4 h”, 5.5 = “5 to 6 h”, 10.5 = “7 to 14 h” and 16 = “15 h or more”). We calculated an overall sum score of PA including all four domains (sports and exercise, active travel, light outdoor PA as well as housework and gardening) for the week preceding the survey and for the pre-pandemic period. PA difference values were then calculated (PA in October 2020—PA in the pre-pandemic period) to estimate changes in PA. The questionnaire for measuring PA adheres to best practice recommendations given by experts [29] and has proven its worth in previous studies (e.g., [19]).

Federal states (place of residence). The place of residence was collected on the Länder level (federal states of Germany). Fourteen Länder are sufficiently represented in the sample with *N* between *N* = 33 (e.g., Hamburg) and *N* = 325 (e.g., Northrhine-Westphalia). We excluded Bremen and the Saarland from our analyses.

COVID-19 policy index. To analyze differences in mitigation policies, major restrictions of each Land were examined based on the official state regulations. We built a policy index that includes seven quantifiable indicators that address different political measures in coping with the pandemic: (1) max. number of people allowed at large events, (2) max. number of people allowed at private celebrations, (3) max. number of people allowed for meetings in public space, (4) hygiene rules at schools, (5) max. number of people allowed for organized sports activities, (6) fines for not wearing a mouth–nose cover, (7) min. required square meters per person in retail shops. We chose these indicators, firstly, because all of them have been the focus of public and political debates. Secondly, all of them may affect the daily routines and risk perceptions of people [30]. Thirdly, there is a considerable variance between the German federal states regarding the strictness of these regulations. These indicators were independently rated by one researcher (FB) and one student assistant. Ratings were made on a 4-point Likert-scale, based on a standardized grading scheme jointly developed by the authors (see Appendix A). Values were then summarized as a policy index for each Land. The index has a value range from 7 to 28, whereby higher values indicate stricter COVID-19 confinement measures (see Table 1).

COVID-19 incidence values. The 7-day incidence per 100,000 inhabitants was collected for every Land. The mean 7-day incidence value between 9 October and 27 October was obtained from the Robert Koch Institute (RKI) [31] (shifted forward by one week compared to the survey period, as the survey questions refer to the last week) (Table 1).

### 2.4. Data Analysis

All statistical tests were conducted using IBM SPSS 26 (IBM Corporation, Armonk, NY, USA). The distribution of the included variables was tested by the Kolmogorov–Smirnov test. Before conducting correlation analysis at the level of Länder, we first calculated the mean PA difference for each Land. To investigate regional differences in PA changes, we then conducted bivariate Pearson correlations between changes in PA levels and the incidence rate as well as PA levels and the COVID-19 policy index. As the included variables were normally distributed, we decided to use a test for Pearson correlation. A level of α = 0.05 was set as a threshold to determine statistical significance.

## 3. Results

Due to the exclusion of Bremen and Saarland, we compared the data from 14 Länder from Germany, including a total of *N* = 1477 participants. The mean 7-day incidence value was 38.7 (SD = 20.44) and ranged from 15.1 to 85.2. The minimum policy index was 12.5 and the maximum was 23.5 (mean = 17.3, SD = 3.37). The changes in PA varied from −1.29 to 0.63 h per week with a mean change of −0.03 (SD = 0.53) hours per week (Table 1 and Appendix A).

Pearson correlations at the level of Länder revealed a strong, negative and significant relationship between changes in PA and the 7-day incidence rate of r = −0.668 (*p* = 0.009) (Figure 1). The correlation between changes in PA and the COVID-19 policy index was moderate and negative but not significant with r = −0.444 (*p* = 0.112) (Figure 2).

## 4. Discussion

The aim of our study was to investigate differences in PA changes related to 7-day incidence rates and mitigation restrictions across the Länder in Germany during the COVID-19 pandemic in October 2020. Results indicated a strong negative correlation between changes in PA and the incidence rates. The Pearson correlation between PA changes and the mitigation policy index revealed a moderate negative association.

Our results showed that the mean change in time spent on PA per week is more negative in Länder with higher incidence values. Stricter policy restrictions that were implemented to slow down the spread of COVID-19 are also associated with more negative changes in PA levels. These findings are in line with a Canadian study focusing on children and adolescents, indicating lower outdoor PA levels in regions with the strictest restrictions resulting from the highest numbers of COVID-19 cases [24]. Nevertheless, the association between COVID-19 policy index and changes in PA level should be viewed with caution, because the correlations can only be classified as marginally significant (*p =* 0.112). This might be explained by the small number of states (*N* = 14).

Our findings align with recent accounts that point to pandemic-related changes in health and health behaviors. With regard to mental health, a German study showed a correlation between higher incidence values and worsening of mental health [23]. Furthermore, a European study investigated associations between mortality and mitigation policies and found lower mortality rates in countries with stricter policies [32]. These findings support the assumption that mitigation policies are associated with health behaviors that may also include PA.

Two mechanisms are likely to explain the correlations found: at an individual level, stricter policy measures and higher incidence rates may increase the salience of the pandemic and may strengthen the impression of its severity. As a consequence, individuals may be more inclined to act cautiously and omit any (physical) activity that may increase the risks of an infection. At a state level, PA opportunities are limited by many mitigation policies. Hence, these policies intentionally constrain the opportunities to be physically active and, for instance, to play sports.

To counteract the decline in PA, it is important to support people in their activity behavior in times of pandemic and restrictions. Politicians, health professionals and public health researchers and practitioners should collaborate to inform people how to stay active during the pandemic. The WHO has already provided guidance on how to stay fit during self-quarantine and how to still achieve the recommendations even at home. Besides general tips like following an online exercise class and walking, the WHO prepared a set of home-based exercises [33].

On the other hand, Länder with lower incidence values and correspondingly more relaxed restrictions (especially new Länder like Saxony, Saxony-Anhalt and Thuringia) showed an increase in overall PA from October 2019 to October 2020. This could be due to a change in the PA levels in different domains and a shift from sports and exercise to daily activities like active transport or exercise in the household or garden. Results from the SPOVID project showed a reduction in sports and exercise levels while PA in the other domains increased [19].

### Limitations

The main strength of this study is the representativeness of the survey participants for the German population related to age, gender, education and place of residence. Furthermore, this is the first study investigating regional differences in PA with regard to mitigation policies and incidence values. Besides the strength of the study, there are some limitations. The self-reported and retrospective data of the survey may have led to recall biases and inaccuracies in PA measures when compared to measures based on accelerometers or pedometers. Furthermore, the SPOVID project is not representative on a Länder level. The sample sizes of the Länder are somewhat small. Furthermore, this small sample size could cause a potential type I error related to the non-significant correlation between PA changes and COVID-19 mitigation policies (*p* = 0.112). A larger sample size might result in a significant relationship. Finally, due to the cross-sectional nature of the survey, we cannot derive causal relationships based on the data.

## 5. Conclusions

This exploratory study aimed to investigate PA changes during the COVID-19 pandemic. These changes were analyzed according to different 7-day incidence values and mitigation policies across the Länder. During the COVID-19 pandemic, changes of PA can be seen across the Länder in Germany. Our results indicated a decrease in PA with increasing incidence values as well as stricter mitigation policies. PA is associated with several health benefits and, thus, following our results, it is important to support individuals to maintain or improve their PA levels in pandemic times. During the pandemic, especially in Länder with high incidence values and strict mitigation policies, measures to promote PA and consequently to improve physical resistance and physical and mental health are desirable.

## Figures and Tables

**Figure 1 sports-09-00102-f001:**
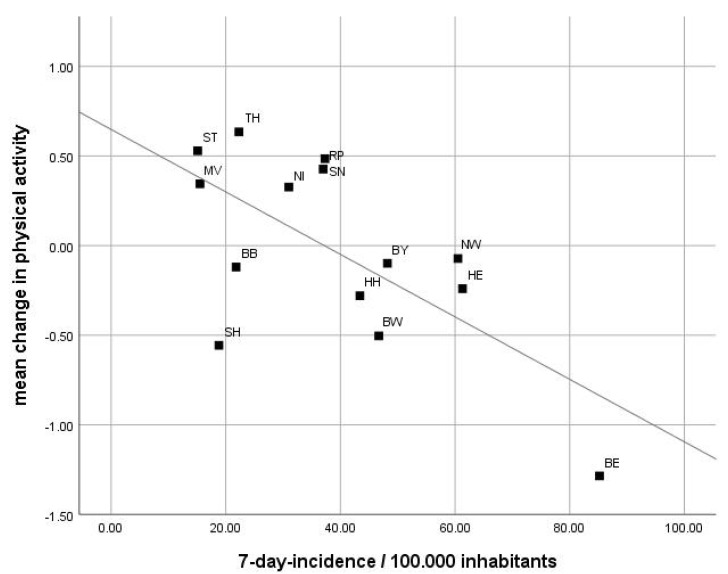
Pearson correlation between mean change in PA and 7-day incidence value with r = −0.688 **, *p* = 0.009. Mean change in time spent on PA per week is more negative in Länder with higher 7-day incidence values.

**Figure 2 sports-09-00102-f002:**
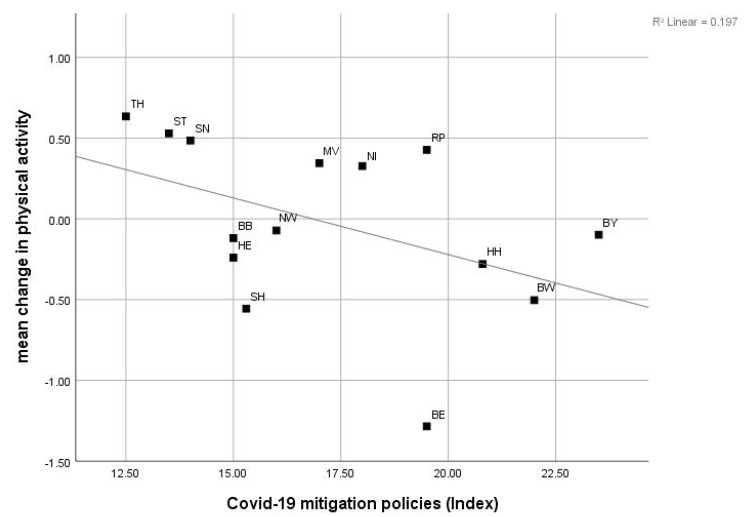
Pearson correlation between mean change in PA and COVID-19 mitigation policy index value with r = −0.444, *p* = 0.112. Mean change in time spent on PA per week is more negative in Länder with higher COVID-19 mitigation policy indexes.

**Table 1 sports-09-00102-t001:** Number of participants, mean 7-day incidence rates per 100,000 inhabitants, policy index and changes in PA for the Länder in Germany.

Länder	Mean 7-Day Incidence Value Per 100,000 Inhabitants	Policy Index	Changes in PA in hrs/Week
Overall/mean	38.7	17.3	−0.03
Berlin (BE)	85.2	19.5	−1.29
Hesse (HE)	61.3	15.0	−0.24
Northrhine-Westphalia (NW)	60.5	16.0	−0.07
Bavaria (BY)	48.2	23.5	−0.10
Baden-Württemberg (BW)	46.7	22.0	−0.50
Hamburg (HH)	43.4	20.8	−0.28
Saxony (SN)	37.3	14.0	0.49
Rhineland Palatinate (RP)	37.0	19.5	0.43
Lower Saxony (NI)	31.0	18.0	0.33
Thuringia (TH)	22.3	12.5	0.63
Brandenburg (BB)	21.8	15.0	−0.12
Schleswig-Holstein (SH)	18.8	15.3	−0.56
Mecklenburg-Western Pomerania (MV)	15.5	17.0	0.34
Saxony-Anhalt (ST)	15.1	13.5	0.53

## Data Availability

The data set of the SPOVID study is available from the first author upon request.

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
