# Peer review of "Changes in Physical Activity during the COVID-19 Pandemic—An Analysis of Differences Based on Mitigation Policies and Incidence Values in the Federal States of Germany"

_sports, 2021, doi:10.3390/sports9070102_

Round 1
Reviewer 1 Report
Please see attachment.

Author Response
Dear Colleague,
Please find enclosed our revision of the following research article:
"Changes in physical activity during the Covid-19 pandemic – an analysis of differences based on mitigation policies and incidence values in the federal states of Germany"
We would like to thank you for your comments. We believe that the quality of our manuscript has improved by implementing the suggestions and comments received. We hope we have addressed all the your concerns and comments to your satisfaction.
Please find our response to the comments in the attached Word file. The reviewers’ comments are shown in regular font, and our responses are shown below each comment in italic.

Reviewer 2 Report
This paper was well-written (needs a few minor commas and other changes here and there) and is important in its content and findings. I had only a few minor comments:
1) page 2 sentence beginning on line 46: I would reword this to be "....sufficiently physically active as World Health Organization guidance recommends".
2) page 2 lines 62-63. This sentence needs rewritten for clarity - I think it is saying out of all German students, 44.5% decreased while 32.8% increased? Also make it clear that these were before any restrictions - or did certain German states begin restrictions in March?
3) on section 2.1 and 2.2 - I would love to have more information about the larger survey such as the total number of questions, other topics examined, etc.
4) Finally, while it is on the verge of .10 significance, it would be good for the authors to discuss explicitly that the p-value for the correlation between mitigation policies and physical activity was small. This is OK to me considering the small N (states).
5) For further consideration, have the authors thought about modeling this in a different fashion (depending on what survey data are available) where you model individual physical activity change using several dependent variables where 7-day incidence and the policy index are two of many variables? This is not necessary for this particular revision, but something to think about to give the sample more power.
Author Response

(The authors gave the same response as above.)

Reviewer 3 Report
Overall, this was a well written manuscript. Below are my comments Introduction The introduction is very well written with literature that is well cited. My one minor comment is that in the second to last paragraph, I would recommend giving some examples of how the mitigation policies are different. Methodology Overall the methodology is well written. Below are my comments 1. For the instrument used to measure PA, can you please provide a citation for validity of the instrument. 2. Was an a priori power analysis completed prior to the study? 3. How was the validity of the COVID-19 policy index measured? Who scored each question? What were the anchors of the Likert scale? The way the measure is currently described it seems very subjective. 4. How was normality of data determined? 5. How did you determine that 2 of the 16 Landers didn't have enough data? What was the reasoning used for that cut-off? 6. Technically with a N of 14 for your bi-variate correlations you should definitely check for normality and you might have to run a non-parametric analysis. 7. What was your alpha? Results Based on the feedback above regarding statistical analyses I don't think your results will change much (you might have larger r's) so I will evaluate the results as if they will not change. Here are my comments. 1. Based on the p-value presented on line 167, is the relationship significant? Unless your alpha is very high, I would recommend not claiming p=.112 as a significant relationship. Discussion Based on the results section I believe there is a huge Type I error thus I cannot evaluate the discussion.Author Response
Dear Colleague,
Please find enclosed our revision of the following research article:
"Changes in physical activity during the Covid-19 pandemic – an analysis of differences based on mitigation policies and incidence values in the federal states of Germany"
We would like to thank you for your comments. We believe that the quality of our manuscript has improved by implementing the suggestions and comments received. We hope we have addressed all the your concerns and comments to your satisfaction.
Please find our response to the comments in the attached Word file. The reviewers’ comments are shown in regular font, and our responses are shown below each comment in italic.

Round 2
Reviewer 3 Report
The authors addressed most of my concerns. The one thing that I would request is that the authors also state in the limitations section the potential for a Type I error as it relates to the second result.
Author Response
Response to article: sports-1268445
Sports
Dear Colleague,
Please find enclosed our revision of the following research article:
"Changes in physical activity during the Covid-19 pandemic – an analysis of differences based on mitigation policies and incidence values in the federal states of Germany"
We would like to thank the reviewer for the comments. We believe that the quality of our manuscript has improved by implementing the suggestions and comments received. We hope we have addressed all the reviewers’ concerns and comments to your satisfaction.
Please find our response to the comment(s) below. The reviewers’ comments are shown in regular font, and our responses are shown below each comment in italic:
Reviewer #3:
The authors addressed most of my concerns. The one thing that I would request is that the authors also state in the limitations section the potential for a Type I error as it relates to the second result.
Thank you for your comment. We added an information about the potential for a Type I error (p.7, line 250-253).
